# Backbone 3.0: An R package for extracting network backbones

**Zachary P. Neal** [ORCID]*

Psychology Department, Michigan State University, East Lansing, Michigan, United States of America

* zpneal@msu.edu

## Abstract

The `backbone` package for R implements models for extracting the backbone – a sparse unweighted network containing only the most 'important' edges – from a weighted or unweighted network, where different models adopt different perspectives on what makes an edge important. The use of network backbones simplifies analysis and visualization when the original network is weighted or dense. This paper introduces and demonstrates the use of the `backbone` package for R to extract network backbones. After providing an overview of `backbone`'s workflow and structure, I describe several backbone extraction models, illustrating backbone extraction in a series of toy examples. I then present a complete empirical case study using bill sponsorship data from the 108th U.S. Senate. I conclude with some recommendations for backbone extraction, and an agenda for planned extensions to `backbone`.

### Extracting network backbones

A network describes the relationships called 'edges' that exist among a set of entities called 'nodes'. For example, a social network might describe the friendships (edges) that exist among a set of people (nodes). When a network has weighted edges (e.g., reflecting the strength of friendship), or when it is very dense (e.g., most people are friends with each other), it can be difficult to visualize and analyze. In such cases, it can be useful to instead focus on the network's *backbone*. The backbone of a network is an unweighted network that retains only the most 'important' edges.

Many models exist for extracting the backbone of a network, with different models adopting different perspectives on what makes an edge important. Statistical backbone extraction models identify important edges by comparing their weights to expected values under a null model. Edges whose weights are statistically significantly larger than expected under a given null model are retained in the backbone. In contrast, structural models identify important edges based on their role in the overall structure of the network using a variety of metrics. Statistical and structural models have been developed for extracting the backbone from several different

**Data availability statement:** The data and code to reproduce all the examples, analyses, and plots are available at https://osf.io/rx6af.

**Funding:** The author(s) received no specific funding for this work.

**Competing interests:** The authors have declared that no competing interests exist.

types of networks: weighted networks, weighted networks formed via projection, and unweighted networks.

The `backbone` package for R [1] implements several backbone extraction models for each of these types of networks. First, it implements four models for extracting the backbone from weighted networks, including the widely-used disparity filter [2]. Second, it implements five models for extracting the backbone from networks whose weights are the product of bipartite or hypergraph projection, including the widely-used stochastic and fixed degree sequence models [3]. Finally, it implements 10 models for extracting the backbone from unweighted networks, including most of the models in a recent review of sparsification methods [4], and offers the option for specifying custom unweighted network backbone models. These models are implemented in a set of functions that are designed to be easy for network researchers to use, but also easy for contributors to extend to future backbone models.

The `backbone` package exists alongside other packages also designed for network backbone extraction or sparsification (see Table 1). Although there is some overlap in the models implemented in `backbone` and these other packages, `backbone` provides greater flexibility as the only package that implements models for multiple types of networks. It is also the only actively-maintained package for R, and thus offers R users an up-to-date option for backbone extraction.

The goal of this article is not to propose or test backbone models, but instead to introduce, demonstrate, and document the `backbone` package as a software that implements backbone models. The remainder of this article is organized in four sections. In the first section, I provide an overview of `backbone`'s workflow and structure. In the second section, I introduce and illustrate the backbone extraction models implemented in `backbone` using toy examples. In the third section, I demonstrate the use of `backbone` with an empirical case study of collaboration in the 108th U.S. Senate. Finally, in the fourth section, I conclude by discussing the limitations of and planned extensions for `backbone`. The code to reproduce all the examples, analyses, and plots shown below is available in S1 and at https://osf.io/rx6af. When software run times are reported, they reflect code execution using a MacBook Pro with an Apple M1 Max processor and 64GB of memory.

## Overview of `backbone`

### Workflow

The `backbone` package can be installed from CRAN and loaded for use in R in the usual way:

```
R> install.packages("backbone")
R> library(backbone)
 ____  backbone v3.0.3
|   \ Cite: Neal, Z. P., (2026). backbone: An R package to extract
|#|_) |  network backbones. CRAN. https://doi.org/10.32614/CRAN
|#  _ <   .package.backbone
|#|_) | Help: type vignette("backbone"); email zpneal@msu.edu
|____/ Beta: type install_github("zpneal/backbone", ref = "devel")
```

**Table 1. Backbone extraction and sparsification models implemented by existing packages.**

| Package | Language | Version | Last Updated | Models | | |
|---|---|---|---|---|---|---|
| | | | | W | P | U |
| backbone [1] | R | 3.0.3 | 2026 | 4 | 5 | 10+ |
| disparityfilter [5] | R | 2.2.3 | 2016 | 1 | — | — |
| simplifyNet [6] | R | 0.0.1 | 2022 | 4 | — | — |
| netbone [7] | python | 0.2.3 | 2024 | 20 | — | — |
| NetworKit [8] | python | 11.1.1 | 2025 | — | — | 8 |

W = weighted network, P = weighted projection of a bipartite network or hypergraph, U = unweighted network

Upon loading, the package startup message displays the version number, citation, and ways to get help. It also shows how to install the package's beta version from GitHub using the `devtools` package [9].

Backbones are extracted using one of three core functions – `backbone_from_weighted()`, `backbone_from_projection()`, or `backbone_from_unweighted()` – depending on the source network type. The source network can be provided to these functions as an incidence or adjacency matrix in a matrix or sparse `Matrix` [10] class object, or as an `igraph` object [11], and by default the resulting backbone is returned as an object of the same class. Each function has two primary arguments: `model` specifies the backbone extraction model, while `alpha` (for statistical models) or `parameter` (for structural models) controls the sparsity of the backbone.

The `backbone()` function is a wrapper that detects the type of source network, and calls the appropriate core function, which provides a convenient user interface:

```
R>dat<- sample_sbm(60,
                matrix(c(.75,.25,.25,.25,.75,.25,.25,.25,.75),3,3),
                c(20,20,20))
R>bb<- backbone(dat)
The backbone package for R (v3.0.3; Neal, 2025) was used to extract the unweighted back-
bone of an unweighted network containing 60 nodes. Edges were selected for retention in
the backbone using Local Sparsification (Satuluri, Parthasarathy, and Ruan, 2011) with
filtering parameter=0.5, which removed 70.4% of the edges.

Neal, Z. P. 2025. backbone: An R Package to Extract Network Backbones. CRAN. https://doi.
org/10.32614/CRAN.package.backbone

Satuluri, V., Parthasarathy, S., & Ruan, Y. (2011, June). Local graph sparsification for
scalable clustering. In Proceedings of the 2011 ACM SIGMOD International Conference on
Management of data (pp. 721-732). https://doi.org/10.1145/1989323.1989399

R> summary(bb)
IGRAPH 294997b U--- 60 194 -- lspar backbone of Stochastic block model
+ attr: name (g/c), loops (g/l), call (g/x), narrative (g/c)
```

For example, given an unweighted igraph object `dat` (here, randomly generated using a stochastic block model), `backbone(dat)` detects that it is an unweighted network and extracts its backbone using the default Local Sparsification model with the default sparsification parameter 0.5. By default, a narrative summary of the process (with citations) is displayed. To allow for an uninterrupted workflow, the result is returned as an igraph object `bb`, which is named to identify the backbone type, and which includes graph attributes containing the function call and a copy of the narrative summary.

Alternatively, specifying `backbone_only=FALSE` returns a `backbone`-class S3 object that contains the original network, backbone network, and additional details. This object can be described using generic `print` and `summary` functions:

```
R> print(bb)
Call -
backbone_from_unweighted(U = N, narrative = narrative,
                 backbone_only=FALSE)
Source: igraph-class unweighted network
Backbone: igraph-class unweighted network
Model: Local Sparsification

R> summary(bb)
NETWORK     TYPE          NODES        EDGES      MIN    MAX
--------------------------------------------------------------
Original  Unweighted     60           763         —      —
Backbone  Unweighted     60           226         —      —
--------------------------------------------------------------
Local Sparsification (parameter = 0.5)
537 (70.4%) edges removed

NARRATIVE SUMMARY -
The backbone package for R (v3.0.3; Neal, 2025) was used to extract the unweighted back-
bone of an unweighted network containing 60 nodes. Edges were selected for retention in
the backbone using Local Sparsification (Satuluri, Parthasarathy, and Ruan, 2011) with
filtering parameter=0.5, which removed 70.4\% of the edges.

Neal, Z. P. 2025. backbone: An R Package to Extract Network Backbones. CRAN. https://doi.
org/10.32614/CRAN.package.backbone

Satuluri, V., Parthasarathy, S., & Ruan, Y. (2011, June). Local graph sparsification for
scalable clustering. In Proceedings of the 2011 ACM SIGMOD International Conference on
Management of data (pp. 721-732). https://doi.org/10.1145/1989323.1989399
```

Additionally, the generic `plot` function returns a side-by-side comparison of the original and backbone networks (see Fig 1).
Although the `backbone()` wrapper may be useful in many cases, except in the concluding empirical case study, the examples in this article use the underlying `backbone_from_*` functions for the sake of being explicit.

### Models

Table 2 summarizes all the backbone extraction models currently implemented in `backbone`, with the relevant calling function and associated `model` argument. This article focuses on the conceptual logic of backbone models, and their practical application using `backbone`. Readers are referred to each model's associated reference for the model's formal mathematical specification and validation.

### Structure

To facilitate package extensibility, the core functions in `backbone` are modular. The statistical models implemented in `backbone_from_weighted()` and `backbone_from_projection()` are performed by first calling an internal function (e.g., `.sdsm()`) to compute edgewise p-values under a specific null model, then calling the internal function `.retain()` to select edges for retention based on these p-values. Similarly, the structural models implemented in `backbone_from_unweighted()` are performed by calling a series of internal functions to assign scores to edges (`.escore()`), to normalize

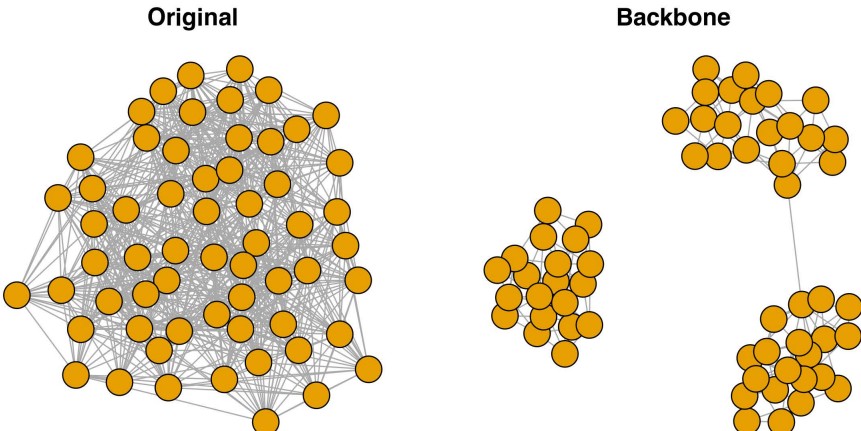

**Fig 1. Comparison of original and backbone networks using S3 plot generic.**

**Table 2. Backbone extraction models implemented in `backbone`.**

| Model | Function/Argument |
|---|---|
| All types of networks | `backbone()` (a wrapper) |
| <u>Weighted networks</u> | `backbone_from_weighted()` |
| Global threshold | `model="global"` |
| Disparity filter [2] | `model="disparity"` |
| Locally adaptive sparsification [12] | `model="lans"` |
| Marginal likelihood filter [13] | `model="mlf"` |
| <u>Weighted projections</u> | `backbone_from_projection()` |
| Stochastic degree sequence model [3] | `model="sdsm"` |
| Fixed degree sequence model [14] | `model="fdsm"` |
| Fixed row [3] | `model="fixedrow"` |
| Fixed column [3] | `model="fixedcol"` |
| Fixed fill [3] | `model="fixedfill"` |
| <u>Unweighted networks</u> | `backbone_from_unweighted()` |
| Local sparsification [15] | `model="lspar"` |
| Global sparsification [15] | `model="gspar"` |
| Local degree [c] | `model="degree"` |
| Skeleton [16] | `model="skeleton"` |
| Simmelian [17] | `model="simmelian"` |
| Simmelian quadrilateral [18] | `model="quadrilateral"` |
| Jaccard [19] | `model="jaccard"` |
| Meetmin [19] | `model="meetmin"` |
| Geometric [19] | `model="geometric"` |
| Hypergeometric [19] | `model="hyper"` |

these scores (`.normalize()`), and to filter edges based on these scores (`.filter()`). This structure simplifies extending backbone to implement new statistical and structural backbone models, which merely requires new or revised internal functions.

To maintain package stability, following the "tinyverse" coding philosophy, dependencies are avoided by using base `R` whenever possible. `backbone` has three first-order dependencies – `igraph` for network manipulation, `Matrix` for sparse

matrix classes, and `Rcpp` for C++ integration [20] – and a total of 12 dependencies. This philosophy has implications for which objects are supported. For example, `backbone` does not support networks stored as `network` objects from the `network` package [21] because doing so would introduce an additional six dependencies. To further maintain package stability, all exported and internal functions are accompanied by unit tests using the `tinytest` package [22].

In addition to the three core functions, two utility functions used by `backbone_from_projection()` are exported because they have potential independent uses. First, `fastball()` is an R wrapper for a fast C++ function that implements the "fastball" algorithm for sampling matrices with fixed marginals [23]. Second, `bicm()` implements the bipartite configuration model for estimating cellwise probabilities in the space of matrices with fixed marginals [24].

## Backbone models illustrated with toy examples

This section provides an introduction to the logic and use of the backbone extraction models, focusing on the most common models for each type of source network. For each model, a small toy network demonstrates the relationship between a source network and its backbone. Network visualizations illustrate these relationships, however in practice network backbones are useful for both visualization and formal analysis.

### Weighted networks

In weighted networks, edges are assigned values that capture the strength of the relationships they represent. Larger values are assumed to represent stronger relationships, but weights may measure strength in different ways (e.g., intensity, frequency) and at different scales (e.g., ordinal, continuous).

```
R>W<- matrix(c(0,10,10,10,10,75,0,0,0,0,
+              10,0,1,1,1,0,0,0,0,0,
+              10,1,0,1,1,0,0,0,0,0,
+              10,1,1,0,1,0,0,0,0,0,
+              10,1,1,1,0,0,0,0,0,0,
+              75,0,0,0,0,0,100,100,100,100,
+              0,0,0,0,0,100,0,10,10,10,
+              0,0,0,0,0,100,10,0,10,10,
+              0,0,0,0,0,100,10,10,0,10,
+              0,0,0,0,0,100,10,10,10,0),10)
R> W <- graph_from_adjacency_matrix(W, mode = "undirected",
                                    weighted=TRUE)
```

In this example, `W` is a weighted network, stored as an igraph object, and plotted in Fig 2A with stronger edges depicted using thicker lines. This network might represent an air traffic system characterized by two hub airports that each serve four regional airports, but where one region and hub carries a much higher volume of passengers than the other.

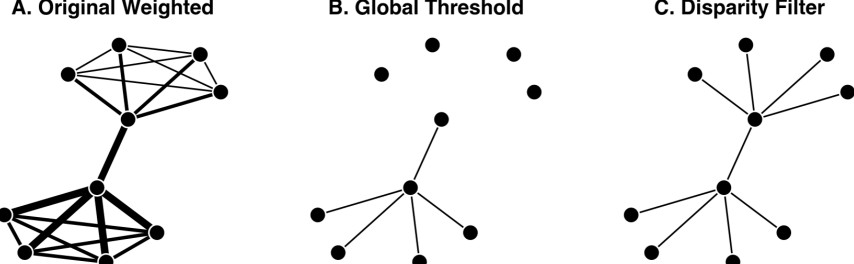

**A. Original Weighted**    **B. Global Threshold**    **C. Disparity Filter**

**Fig 2. Extracting a backbone from a weighted network. (A)** A toy weighted network with a multi-scale hub-and-spoke structure, where stronger edges are represented by thicker lines. **(B)** A backbone extracted from (A) using the mean edge weight as a global threshold, which only preserves high-weight edges. **(C)** A backbone extracted from (A) using the disparity filter, which preserves the hub-and-spoke structure.

We can extract backbones from weighted networks using the `backbone_from_weighted()` function. This function implements one structural model, the *global threshold*, which simply retains all edges whose weights are larger than a specified value.

```
R>bb<- backbone_from_weighted(W,
+           model = "global",
+           parameter = mean(E(W)$weight))
```

Here, we extract a global threshold backbone using the mean edge weight as the threshold value. This yields a backbone returned as an igraph object that contains only edges whose weights were larger than average (see Fig 2B). Because the global threshold model applies the same threshold value to all edges, it ignores the multiscale features of this network. In this case, it yields a backbone that preserves edges in the high-volume region, but not in the low-volume region, which fails to capture the network's underlying structure.

The `backbone_from_weighted()` function also implements three statistical models: *disparity filter* [2], *locally adaptive network sparsification* [12], and *marginal likelihood filter* [13]. These models retain edges whose weights are statistically significantly larger than expected under a null model at a given $\alpha$ level.

```
R>bb<- backbone_from_weighted(W,
+           model = "disparity",
+           alpha = 0.05)
```

Here, we extract the disparity filter backbone at the $\alpha$ = 0.05 level of statistical significance. By evaluating each edge's weight against its expected value in a null model that considers local variations in edge weights, this yields a backbone that preserves some edges with larger weights and some edges with smaller weights, and thereby captures the network's underlying hub-and-spoke structure (see Fig 2C).

Locally adaptive network sparsification (using `model = "lans"`) and the marginal likelihood filter (using `model = "mlf"`) yield the same backbone as the disparity filter in this example. In general, all three models implemented in `backbone_from_weighted()` — `disparity`, `lans`, and `mlf` — yield similar backbones and have similar runtimes. Therefore, while any of these models may be appropriate, because it is the most highly-cited and widely-known, the disparity filter is often the better choice and is the default. By implementing multiple models for extracting the backbone from weighted networks, `backbone` facilitates further research on their similarities and differences [7].

## Weighted projections

The backbone models available using `backbone_from_weighted()` can be applied to any weighted network. However, specialized models exist for weighted networks that were obtained by projecting a bipartite network or hypergraph (i.e., where the edge weights represent co-occurrences).

```
R>B<- rbind(cbind(matrix(rbinom(250,1,.8),10),
+                 matrix(rbinom(250,1,.2),10),
+                 matrix(rbinom(250,1,.2),10)),
+           cbind(matrix(rbinom(250,1,.2),10),
+                 matrix(rbinom(250,1,.8),10),
+                 matrix(rbinom(250,1,.2),10)),
+           cbind(matrix(rbinom(250,1,.2),10),
+                 matrix(rbinom(250,1,.2),10),
+                 matrix(rbinom(250,1,.8),10)))
R>B<- graph_from_biadjacency_matrix(B)
```

In this example, `B` is a bipartite network, stored as an igraph object, and plotted in Fig 3A. In generic terms, this network contains 30 *agents* connected to 75 *artifacts*. It might represent 30 researchers' and 75 articles, where a researcher is connected to an article if they were an author. These toy data are constructed to contain three embedded communities. This might represent a case where the researchers and articles each belong to one of three disciplines, such that researchers are more likely to have authored a paper inside their discipline than outside their discipline. Bipartite networks are challenging to analyze, so analysis often focuses on their unipartite projection [25].

```
P <- bipartite_projection(B, which = "true")
```

`P` is the weighted unipartite projection of `B` on the agent nodes, and is plotted in Fig 3B. In this weighted projection, two agents are connected if they shared artifacts, and the weight of the edge between them captures their number of shared artifacts. For example, this projection might capture patterns of co-authorship among researchers. However, these weights are "noisy" because they are affected by the agents' degrees and artifacts' degrees (i.e., the marginals of `B`) [26]. Despite the fact that these data are known to contain three communities, this community structure is difficult to detect in the weighted projection.

## A. Original Bipartite

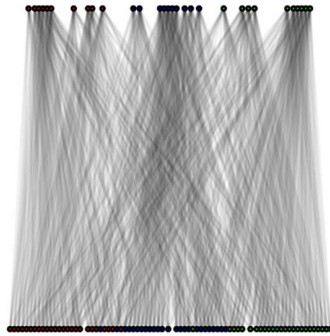

## B. Weighted Projection

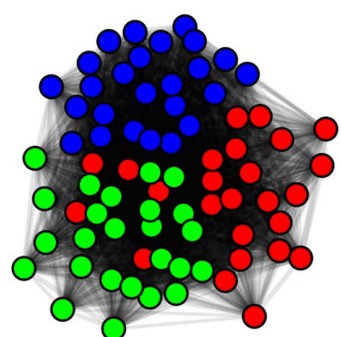

## C. SDSM Backbone

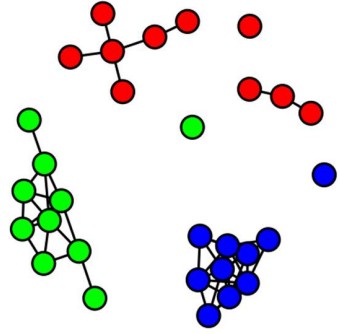

## D. FDSM Backbone

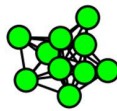
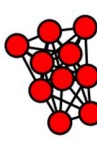
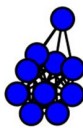

**Fig 3. Extracting a backbone from a weighted projection. (A)** A toy bipartite network with three embedded communities. **(B)** The weighted projection of **(A)**, where stronger edges are represented by thicker lines. **(C)** A backbone extracted from (B) using the stochastic degree sequence model (SDSM), which correctly preserves the known three communities. **(D)** A backbone extracted from (B) using the fixed degree sequence model (FDSM), which correctly preserves the known three communities.

The `backbone_from_bipartite()` function implements five statistical models for extracting the backbone from weighted projections: *stochastic degree sequence model* (SDSM), *fixed row model*, *fixed column model*, *fixed fill model* [3] and *fixed degree sequence model* (FDSM) [14]. These models differ in the constraints they impose on `B` in the null model used to evaluate the statistical significance of edge weights in the projection. For example, the fixed row model uses a null model that constrains the row marginals of `B`, while the fixed column model uses a null model that constrains the column marginals of `B`.

```
R>bb1<- backbone_from_projection(B,
+        model = "sdsm",
+        alpha = 0.05)
R>bb2<- backbone_from_projection(B,
+        model = "fdsm",
+        alpha = 0.05)
```

Here, we extract the backbone of the weighted projection using the SDSM (using `model="sdsm"`) and FDSM (using `model="fdsm"`) at the $\alpha$ = 0.05 level of statistical significance, which are plotted in Figs 3C and D, respectively. Importantly, although the backbone models implemented by `backbone_from_projection()` are designed to extract the backbone from a weighted projection, they operate on the original bipartite network or hypergraph from which the projection was derived. Thus, the source network in these functions is `B`, and not `P`.

These two backbones are similar, and both reveal the three-community structure that is known to exist in these data. The null model used by the SDSM is computationally efficient but statistically less powerful; it takes only 0.039 seconds to run, but detects fewer statistically significant edges. Conversely, the null model used by the FDSM is computationally intensive but statistically more powerful; it takes 13.131 seconds to run (over 300 times longer), but detects more statistically significant edges. The tradeoff of efficiency for power is the primary consideration in choosing between SDSM and FDSM as a model for extracting the backbone from a weighted projection. Given FDSM's potentially long runtime, SDSM is often the better choice and is the default. The other models implemented in `backbone_from_projection()` — `fixedrow`, `fixedcol`, and `fixedfill` — are known to perform worse and generally should not be used. They are implemented in `backbone` for their methodological, rather than practical, applications [3].

## Unweighted networks

Extracting backbones from unweighted networks is more challenging because the network does not explicitly contain weight information that can be used to judge which edges are more important. Instead, the `backbone_from_unweighted()` function extracts backbones from unweighted networks by following a series of steps:

1. Assign each edge a score using a structural metric specified by `escore`.

2. Optionally normalize these scores using a method specified by `normalize`.

3. Filter the edges based on these scores using a method specified by `filter`.

4. Optionally include edges in the union of maximum spanning trees to ensure the backbone is connected, specified by `umst`.

Each combination of these four arguments to `backbone_from_unweighted()` specifies a unique backbone extraction model. However, some combinations specify backbone models that have previously been described in the literature, and can be specified directly using `model`. In this section, I illustrate two such models that are designed to preserve different structural properties in the backbone.

```
R>U<- sample_sbm(60,
+      matrix(c(.75,.25,.25,.25,.75,.25,.25,.25,.75),3,3),
+      c(20,20,20))
R>bb<- backbone_from_unweighted(U,
+       model = "lspar",
+       parameter = 0.5)
```

In this example, `U` is an unweighted network constructed using a stochastic block model, stored as an igraph object, and plotted in Fig 4A. It is constructed to contain three embedded communities, but they are obscured by the high density. We can extract the backbone from this network using *Local Sparsification*(L-Spar) [15] by specifying `model="lspar."` It is equivalent to weighting edges by the jaccard coefficient of their endpoints' neighborhoods (`escore="jaccard"`), normalizing these scores by ranking them locally (`normalize="rank"`), filtering them from the perspective of each node's degree (`filter="degree"`), and not including the union of maximum spanning trees (`umst=FALSE`). This backbone reveals the three communities that are known to exist (see Fig 4B).

```
R> U <- sample_pa(n = 60,
+       m = 3,
+       directed = FALSE)
R>bb<- backbone_from_unweighted(U,
+       model = "degree",
+       parameter = 0.25)
```

## A. Original w/ communities
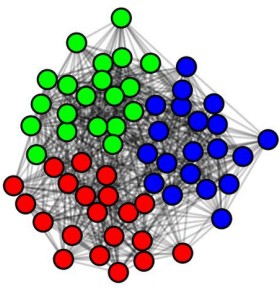

## B. L-Spar Backbone
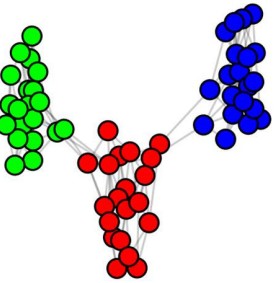

## C. Original w/ hubs
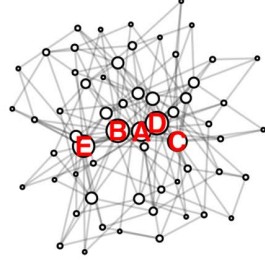

## D. Local Degree Backbone
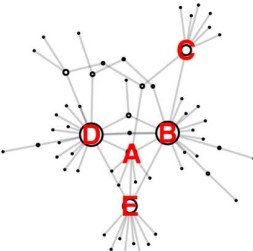

**Fig 4. Extracting backbones from unweighted networks. (A)** An toy unweighted network with three embedded communities. **(B)** A backbone extracted from (A) using Local Sparsification (L-Spar), which preserves the three communities. **(C)** A toy unweighted network with embedded high-degree hubs. **(D)** A backbone extracted from (C) using Local Degree, which preserves the high-degree hubs.

In this example, `U` is an unweighted network constructed using a preferential attachment model, stored as an igraph object, and plotted in Fig 4C. It is constructed to contain high-degree hubs (the five highest are labeled), but they are obscured by the high density. We can extract the backbone from this network using the *Local Degree* model [4] by specifying `model = "degree."` It is equivalent to weighting edges by the degree of their neighbors (`escore = "degree"`), normalizing these scores by ranking them locally (`normalize = "rank"`), filtering them from the perspective of each node's degree (`filter = "degree"`), and not including the union of maximum spanning trees (`umst = FALSE`). This backbone reveals the high-degree hubs and preserves their rank in the degree distribution (see Fig 4D).

The `backbone_from_unweighted()` function implements an additional eight backbone extraction models: (1) *skeleton* [16], (2) *global sparsification* [15], (3) *Simmelian* [17], (4) *Quadrilateral Simmelian* [18], and (567–8) the *jaccard, meetmin, geometric*, and *hypergeometric* backbones [19]. When the goal is to preserve or uncover hidden clusters or communities, `lspar` is often the better option and is the default, but `gspar`, `simmelian`, `quadrilateral`, `jaccard`, `meetmin`, `geometric`, and `hyper` often yield similar results. When the goal is to preserve or uncover hidden hubs or high-degree nodes, `degree` is often the better option and is the default. Using `skeleton` is not recommended because it yields non-reproducible random backbones; it is implemented for its methodological, rather than practical, application. By implementing multiple models for extracting the backbone from unweighted networks, `backbone` facilitates further research on their similarities and differences [4].

## Empirical example: Bill sponsorship in 108th U.S. Senate

### Data

In this section, I demonstrate the use of `backbone` in the empirical context of understanding political collaboration.

```
R> data(senate108)
R> summary(senate108)
IGRAPH 42f8dc2 UN-B 3135 19060 --
+ attr: name (v/c), type (v/l), party (v/c), state (v/c), last (v/c),
id (v/c), color (v/c), introduced (v/c), title (v/c), area (v/c),
sponsor.party (v/c), partisan (v/n), status (v/c)
```

The igraph object `senate108` is bundled with `backbone` and contains data on bill sponsorship in the U.S. Senate between 3 January 2003 and 3 January 2005 (the 108th session) in the form of a bipartite network of 100 Senators and 3035 bills, connected by 19,060 edges representing bill sponsorship. The object also includes several node attributes for both the Senators (e.g., name, party) and bills (e.g., title, status).

```
R> as_biadjacency_matrix(senate108)[1:5,1:5]
                           S144  S1612 S1379 S504 S1709
Sen. Akaka, Daniel K. [D-HI] 1     1     1     1     1
Sen. Allard, Wayne [R-CO]    0     0     0     0     0
Sen. Allen, George [R-VA]    0     0     1     0     0
Sen. Alexander, Lamar [R-TN] 0     0     0     1     0
Sen. Baucus, Max [D-MT]      1     0     1     0     0
R> degree(senate108)[1:5]
   AKAKA   ALLARD    ALLEN  Alexander  BAUCUS
     188      120      203         99     188
R> degree(senate108)[101:105]
S144  S1612 S1379 S504 S1709
13     11    76    38    20
```

As the incidence matrix illustrates, Senators are connected to the bills they sponsored or co-sponsored (e.g., Sen. Allen sponsored or co-sponsored S1379, the "American Veterans Disabled for Life Commemorative Coin Act"). The degrees of

the Senators illustrate that some Senators sponsored more bills than others (e.g., Allen sponsored twice as many as Alexander), and that some bills were more popular than others (e.g., S1379 was sponsored by nearly 7 times more Senators than S1612).

```
R> P <- bipartite_projection(senate108, which = "false")
R> as_adjacency_matrix(P, attr = "weight", sparse = FALSE)[1:5,1:5]
          AKAKA  ALLARD  ALLEN  Alexander  BAUCUS
AKAKA        0     14     38        19       37
ALLARD      14      0     41        22       24
ALLEN       38     41      0        37       31
Alexander   19     22     37         0       17
BAUCUS      37     24     31        17        0
```

The weighted projection of this bipartite network captures pairs of Senators who sponsored the same bills, weighted by the number of bills they both sponsored, and may provide insight into patterns of collaboration or ideological alignment in the chamber. For example, the adjacency matrix of the projection illustrates that Senators Allard and Allan (both Republicans) sponsored 41 bills together, while Senators Allard and Akaka (members of different parties) sponsored only 14 bills together. However, these edge weights are noisy because they are heavily influenced by the Senators' and bills' degrees [26].

Depending on the data we have available, we can use different backbone extraction methods to uncover hidden structure. In the sections below, I illustrate how different backbone extraction methods might be used under conditions where different amounts of data are available (e.g., SDSM if the bipartite data is available, local sparsification if only a binary version of the projection is available). This illustration also highlights that all relevant backbone extraction methods are computationally fast, with each one requiring less than 1 second of computing time.

### Backbone from the weighted projection using SDSM

```
R>bb<- backbone(senate108)
```

```
The backbone package for R (v3.0.3; Neal, 2025) was used to extract the unweighted back-
bone of the weighted projection of a bipartite network containing 100 agents and 3035
artifacts. An edge was retained in the backbone if its weight was statistically signifi-
cant (alpha=0.05) using the stochastic degree sequence model (SDSM; Neal, Domagalski, and
Sagan, 2021), which reduced the number of edges by 82.93%
```

```
Neal, Z. P. 2025. backbone: An R Package to Extract Network Backbones. CRAN. https://doi.
org/10.32614/CRAN.package.backbone
```

```
Neal, Z. P., Domagalski, R., and Sagan, B. (2021). Comparing Alternatives to the Fixed
Degree Sequence Model for Extracting the Backbone of Bipartite Projections. Scientific
Reports, 11, 23929. https://doi.org/10.1038/s41598-021-03238-3
```

The wrapper function `backbone()` detects that `senate108` is a bipartite network (see Figure 5A1), and calls `backbone_from_projection()` to extract the backbone from its weighted projection using the default SDSM model at the default $\alpha$ = 0.05 level of statistical significance. By default, the wrapper function displays narrative text describing what it did. This function requires 0.336 seconds to run.

**If you have...**  **You can extract...**

**A1. A Bipartite network**  **A2. An SDSM Backbone**

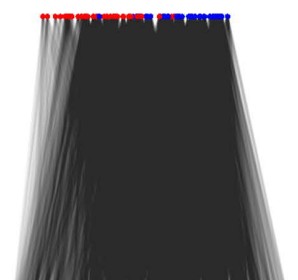 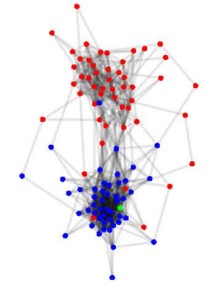

**B1. A Weighted Network**  **B2. A Disparity Filter Backbone**

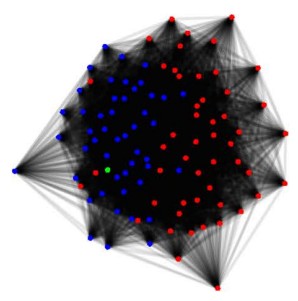 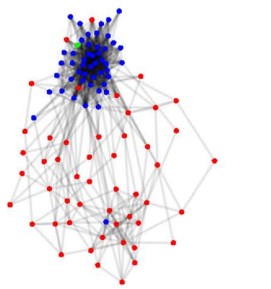

**C1. An Unweighted Network**  **C2. An L-Spar Backbone**

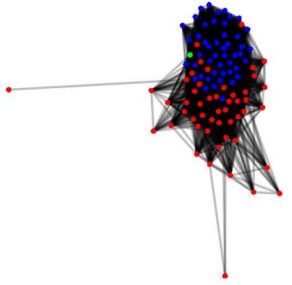 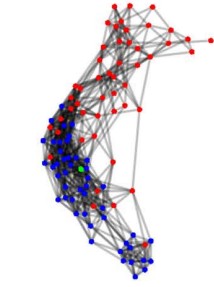

**Fig 5. Backbones of bill co-sponsorship in the 108th U.S. Senate, where nodes representing Republican Senators are red and nodes representing Democratic Senators are blue.** If the source data represent a bipartite network (A1), then a backbone of the weighted projection can be extracted using a model such as the stochastic degree sequence model (SDSM, A2). If the source data represent a weighted network (B1), then a backbone can be extracted using a model such as the disparity filter (B2). If the source data represent an unweighted network, then a backbone can be extracted using a model such as local sparsification (L-Spar, C2). All backbones correctly capture the Senate's partisan polarization, where Senators' collaborations are clustered by party affiliation.

This yields a backbone (see Figure 5A2) that clearly captures the partisan polarization known to structure interactions in the U.S. Senate. Republicans (red nodes) primarily collaborate with other Republicans, and Democrats (blue nodes) primarily collaborate with other Democrats. Additionally, it illustrates that there are a few bipartisan Senators (nodes bridging the two communities), as well as a few conservative-leaning Democrats (blue nodes in the red community) and liberal-leaning Republicans (red nodes in the blue community).

### Backbone from the weighted projection using disparity filter

```
R> bb <- backbone(P, alpha = 0.2)
```

```
The backbone package for R (v3.0.3; Neal, 2025) was used to extract the unweighted back-
bone of a weighted network containing 100 nodes. An edge was retained in the backbone if
its weight was statistically significant (alpha=0.2) using the disparity filter (Serrano
et al., 2009), which reduced the number of edges by 83.29%
```

Neal, Z. P. 2025. backbone: An R Package to Extract Network Backbones. CRAN. https://doi.org/10.32614/CRAN.package.backbone

Serrano, M. A., Boguna, M., & Vespignani, A. (2009). Extracting the multiscale backbone of complex weighted networks. Proceedings of the National Academy of Sciences, 106, 6483–6488. https://doi.org/10.1073/pnas.0808904106

However, suppose we did not have access to the original bipartite data, and instead only had access to the weighted projection P shown in Figure 5B1. Here, the `backbone()` function detects that P is a weighted network, and calls `backbone_from_weighted()` to extract its backbone using the default Disparity Filter model. We supply the optional `alpha=0.2` to specify a more liberal $\alpha$ = 0.2 level of statistical significance to account for the fact that the weighted projection contains less information than the original bipartite. This function runs in 0.01 seconds.

This yields a backbone (see Figure 5B2) that still captures the partisan polarization. Here, it is reflected in a dense community of collaborating Democrats, and a more diffuse community of collaborating Republicans. In the 108th U.S. Senate, Republicans were in the majority. Thus, this pattern is consistent with prior findings that members of the majority can stake out more extreme positions and exhibit "strategic disloyalty," while members of the minority party must work more closely together to maximize their power [27, 28].

### Backbone from an unweighted projection using Local Sparsification

```
R>bb<- backbone(U)
```

```
The backbone package for R (v3.0.3; Neal, 2025) was used to extract the unweighted back-
bone of an unweighted network containing 98 nodes. Edges were selected for retention in
the backbone using Satuluri et al's (2011) Local Sparsification backbone model (filtering
parameter=0.5), which reduced the number of edges by 79.68%
```

Neal, Z. P. 2025. backbone: An R Package to Extract Network Backbones. CRAN. https://doi.org/10.32614/CRAN.package.backbone

Satuluri, V., Parthasarathy, S., & Ruan, Y. (2011, June). Local graph sparsification for scalable clustering. In Proceedings of the 2011 ACM SIGMOD International Conference on Management of data (pp. 721–732). https://doi.org/10.1145/1989323.1989399

However, suppose we did not have access to the weighted projection, and instead only had access to a simple unweighted projection in which Senators are connected if they sponsored at least 25 of the same bills (see Figure 5C1). Here, the `backbone()` function detects that `U` is an unweighted network, and calls `backbone_from_unweighted()` to extract its backbone using the default Local Sparsification model with the default sparsification parameter of 0.5. This function runs in 0.007 seconds. Even based on this very limited data, it still yields a backbone (see Figure 5C2) that captures partisan polarization, again with a higher density of collaboration among members of the minority party.

## Comparing models and empirical results

Table 3 summarizes the backbone models used in the empirical case of studying networks of legislative collaboration. These three models – the stochastic degree sequence model (SDSM), disparity filter, and local sparsification (L-Spar) – share many common features: They all return an unweighted network (i.e., the backbone) in less than one second that correctly captures the partisan polarization known to exist in the U.S. Senate.

However, they also differ in important ways. First, each model requires a different type of input or starting network. The SDSM has the most stringent input requirement (the complete bipartite network from which a projection was obtained; here, which Senators sponsored which bills), while the L-Spar has the weakest input requirement (an unweighted, unipartite network; here, which Senators sponsored more than 25 bills together). Second, these different input requirements also mean that the models return backbones based on different amounts of information. The SDSM returns a backbone based on more information (here, information about *both* Senators and bills), while the L-Spar returns a backbone based on relatively little information (here, censored information about Senators only). Third, the parameter that controls these models differs in its interpretability. As statistical models, the $\alpha$ parameter used by the SDSM and disparity filter has a formal statistical interpretation and can be selected following disciplinary statistical conventions. In contrast, as a structural model, the parameter used by the L-Spar has no direct interpretation and the selection of a particular value is arbitrary.

This empirical case study illustrates that the data available to the researcher shapes the choice of backbone model; the SDSM requires more data (a full bipartite network) while the L-Spar requires less data (only an unweighted network). It also illustrates that even when limited data is available, the backbone can still reveal or preserve key patterns. However, in practice researchers should extract a backbone from the most information-rich input available. For example, if a bipartite network is available, the backbone should be extracted using SDSM, and should not be extracted from a weighted projection using disparity filter or from an unweighted projection using L-Spar.

**Table 3. Comparison of models and empirical results.**

|  | Stochastic Degree Sequence Model | Disparity Filter | Local Sparsification |
|---|---|---|---|
| **Output** | Unweighted network | Unweighted network | Unweighted network |
| **Time**[a] | 0.336 | 0.01 | 0.007 |
| **Pattern** | Partisan polarization | Partisan polarization | Partisan polarization |
| **Input** | Bipartite network | Weighted network | Unweighted network |
| **Uses** | Information about Senators and bills | Information about Senators | Censored information about Senators |
| **Parameter** | Has formal statistical interpretation | Has formal statistical interpretation | Selection is arbitrary, no interpretation |

[a] Running time in seconds

## Advanced backbone extraction options

Whether extracting a backbone from a weighted projection with `backbone_from_projection()` or from a weighted network with `backbone_from_weighted()` using a statistical model, there are two ways that we might consider refining our backbone: multiple test corrections and signed backbones.

Statistical models involve performing an independent test to evaluate each edge, which can lead to an inflated Type-I error rate. We can adjust for this by applying a multiple test correction using the `mtc` argument, which accepts any method implemented in `R`'s `p.adjust` function. These corrections yield sparser backbones because they involve more stringent statistical tests. However, extracting a backbone that applies the false discovery rate (FDR) [29] correction by specifying `mtc="fdr"` still yields a backbone that captures the polarized structure (see Fig 6A).

By default, statistical models aim to detect edges whose weights are statistically significantly *stronger* than expected under a null model using a one-tailed test. However, we can also detect edges whose weights are statistically significantly *weaker* than expected under a two-tailed test with `signed=TRUE`. Including this option returns a signed backbone in which strong edges are retained as positive and weak edges are retained as negative. Fig 6B plots the signed SDSM backbone extracted at the $\alpha$ = 0.1 level of statistical significance, with positive edges shown in green and negative edges shown in red. This backbone captures the within-party collaborations that were visible in Fig 5B as positive edges, but also contains between-party oppositions or ideological misalignments as negative edges.

## Discussion

The `backbone` package for `R` implements multiple network backbone extraction models for use with multiple types of networks that are summarized in Table 2. It therefore offers network researchers with a range of options for simplifying dense or weighted networks to facilitate their analysis or visualization. This article has demonstrated the logic and use of these models in a series of toy examples, and has demonstrated their empirical utility in the context of understanding collaboration in the U.S. Senate. In this section, I conclude by discussing three directions for the future development of `backbone`.

First, the literature is filled with backbone models, and new models are continuously being proposed. Although `backbone` implements the most widely-used and highly-cited models, it is structured to facilitate the addition of new models. Specifically, each model is implemented as an independent internal function (e.g., `.disparity()`) that is called by the

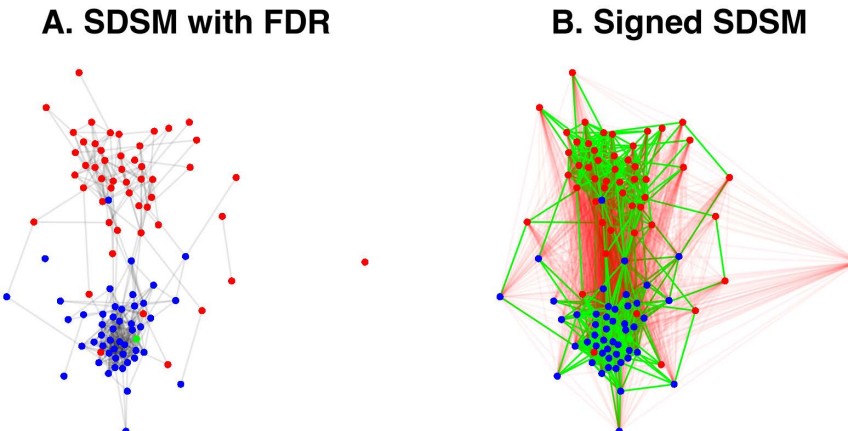

**A. SDSM with FDR**  **B. Signed SDSM**

**Fig 6. Advanced backbones of bill co-sponsorship in the 108th U.S. Senate, where nodes representing Republican Senators are red and nodes representing Democratic Senators are blue. (A)** A backbone extracted using the stochastic degree sequence model (SDSM), correcting for multiple tests by controlling the false discovery rate (FDR) at 0.05. **(B)** A signed backbone extracted using the stochastic degree sequence model (SDSM), where statistically significantly strong edges are preserved as positive (green) and statistically significantly weak edges are preserved as negative (red).

appropriate `backbone_from_` function. Thus, new models can easily be added through open-source development as contributors prepare new internal functions that follow the style of existing internal functions.

Second, although most of the implemented backbone models run quickly even for networks containing hundreds or thousands of nodes, there exist opportunities to improve the computational efficiency of the implementations. Significant gains in efficiency can be achieved in future versions by translating internal model functions into `C++`, which can be called by `backbone` using `Rcpp` [20]. The exported `fastball()` function used by the internal `.fdsm()` function to randomize incidence matrices uses this approach, and serves as a template.

Finally, despite a large literature on methods of network backbone extraction and sparsification, there are relatively few formal comparisons of these methods [7, 3, 4]. However, understanding how different backbone extraction models relate to one another is essential for refining existing models and guiding the development of new models. It is also essential for providing network researchers with guidance on model selection. By implementing multiple backbone extraction methods in a common and extensible framework, `backbone` facilitates this research.

## Supporting information

**S1 File. Backbone.R** This file contains the R code necessary to reproduce all examples in this manuscript. It is also available in the associated repository at https://osf.io/rx6af.
(R)

## Author contributions

**Conceptualization:** Zachary P. Neal.

**Data curation:** Zachary P. Neal.

**Formal analysis:** Zachary P. Neal.

**Investigation:** Zachary P. Neal.

**Methodology:** Zachary P. Neal.

**Software:** Zachary P. Neal.

**Visualization:** Zachary P. Neal.

**Writing – original draft:** Zachary P. Neal.

**Writing – review & editing:** Zachary P. Neal.

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
