## [Decision Letter · Decision Letter 0]

8 Apr 2026

PONE-D-26-02567backbone 3.0: An R package for extracting network backbonesPLOS One

Dear Dr. Neal,

Thank you for submitting your manuscript to PLOS ONE. After careful consideration, we feel that it has merit but does not fully meet PLOS ONE’s publication criteria as it currently stands. Therefore, we invite you to submit a revised version of the manuscript that addresses the points raised during the review process.

We look forward to receiving your revised manuscript.

Kind regards,

Ye Wei, PhD

Academic Editor

PLOS One

3. We are unable to open your Supporting Information file [backbone3.R]. Please kindly revise as necessary and re-upload.

Reviewers' comments:

Reviewer's Responses to Questions

**Comments to the Author**

1. Is the manuscript technically sound, and do the data support the conclusions?

Reviewer #1: Yes

Reviewer #2: Yes

2. Has the statistical analysis been performed appropriately and rigorously?

Reviewer #1: Yes

Reviewer #2: Yes

3. Have the authors made all data underlying the findings in their manuscript fully available?

Reviewer #1: Yes

Reviewer #2: Yes

4. Is the manuscript presented in an intelligible fashion and written in standard English?

Reviewer #1: Yes

Reviewer #2: Yes

5. Review Comments to the Author

Reviewer #1: This manuscript presents backbone 3.0, an R package for extracting network backbones from weighted, bipartite projection, and unweighted networks. The paper is technically sound and demonstrates the package's functionality through well-designed examples. However, in its current form, the manuscript lacks sufficient critical engagement with the broader literature on network backbone extraction and sparsification methods.

While the paper provides detailed technical descriptions of the implemented models, it does not adequately discuss the relative strengths and weaknesses of different backbone extraction models when applied to the empirical network of bill sponsorship from the 108th U.S. Senate. This omission significantly limits the manuscript's utility for practitioners who must select appropriate methods for their specific research contexts. When applying these methods to the empirical case, a summary table is suggested to explicitly indicate: (a) whether each model is statistical or structural, (b) computational complexity, (c) its strengths in capturing what kind of patterns or structures, and (d) key limitations. In addition, the network figure is not that self-explained and the patterns are a bit messy.

I recommend minor revision to address these concerns, after which the manuscript should be suitable for publication.

Reviewer #2: This manuscript introduces backbone 3.0, an R package for extracting network backbones from weighted, projected, and unweighted networks. The paper clearly demonstrates the functionality of the package through illustrative examples and an empirical case study based on U.S. Senate data. Overall, the work has practical value, particularly in providing a unified and actively maintained tool within the R ecosystem.

However, the manuscript would benefit from a more thorough discussion of model selection. Although multiple backbone extraction methods are implemented, there is no systematic comparison of their performance under different network conditions. As a result, it remains unclear when a particular model should be preferred. Including simulation experiments (e.g., varying network density or noise levels) and quantitative comparisons (such as structure recovery or stability) would significantly strengthen the paper and provide useful guidance to users.

In addition, the role of key parameters (e.g., alpha) is not sufficiently discussed. The sensitivity of results to these parameters is unclear, which may make it difficult for practitioners—especially beginners—to apply the methods appropriately. Providing some guidance or empirical analysis on parameter selection would improve usability.

Finally, the computational efficiency of different models is not clearly evaluated. Since some methods may differ substantially in runtime, it would be helpful to include basic performance comparisons to inform practical use.

Overall, the manuscript is useful but would benefit from these clarifications and additional analyses.

6. PLOS authors have the option to publish the peer review history of their article (what does this mean?). If published, this will include your full peer review and any attached files.

Reviewer #1: No

Reviewer #2: No

---

## [Author Response · Author response to Decision Letter 1]

14 Apr 2026

Please see attached "Response to Reviewers.pdf" file.

---

## [Decision Letter · Decision Letter 1]

27 Apr 2026

backbone 3.0: An R package for extracting network backbones

PONE-D-26-02567R1

Dear Dr. Neal,

We’re pleased to inform you that your manuscript has been judged scientifically suitable for publication and will be formally accepted for publication once it meets all outstanding technical requirements.

Kind regards,

Ye Wei, PhD

Academic Editor

PLOS One

Additional Editor Comments (optional):

Reviewers' comments:

Reviewer's Responses to Questions

**Comments to the Author**

1. If the authors have adequately addressed your comments raised in a previous round of review and you feel that this manuscript is now acceptable for publication, you may indicate that here to bypass the “Comments to the Author” section, enter your conflict of interest statement in the “Confidential to Editor” section, and submit your "Accept" recommendation.

Reviewer #1: All comments have been addressed

Reviewer #2: All comments have been addressed

2. Is the manuscript technically sound, and do the data support the conclusions?

Reviewer #1: Yes

Reviewer #2: Yes

3. Has the statistical analysis been performed appropriately and rigorously?

Reviewer #1: Yes

Reviewer #2: Yes

4. Have the authors made all data underlying the findings in their manuscript fully available?

Reviewer #1: Yes

Reviewer #2: Yes

5. Is the manuscript presented in an intelligible fashion and written in standard English?

Reviewer #1: Yes

Reviewer #2: Yes

6. Review Comments to the Author

Reviewer #1: I am satisfied with the revision and have no further comments. I believe the paper now meets the standards of Plos One and I am pleased to recommend it for publication.

Reviewer #2: The author has fully addressed all of my concerns, and I agree to accept the manuscript for publication.

7. PLOS authors have the option to publish the peer review history of their article (what does this mean?). If published, this will include your full peer review and any attached files.

Reviewer #1: No

Reviewer #2: **Yes:** weiyang zhang

---

## [Editor Report · Acceptance letter]

PONE-D-26-02567R1

PLOS One

Dear Dr. Neal,

I'm pleased to inform you that your manuscript has been deemed suitable for publication in PLOS One. Congratulations! Your manuscript is now being handed over to our production team.

Kind regards,

on behalf of

Dr. Ye Wei

Academic Editor

PLOS One